# Efficient Bioprocess for Mixed PET Waste Depolymerization Using Crude Cutinase

**DOI:** 10.3390/polym17060763

**Published:** 2025-03-14

**Authors:** Virender Kumar, Reinhard Wimmer, Cristiano Varrone

**Affiliations:** Department of Chemistry and Bioscience, Aalborg University, Fredrik Bajers Vej 7H, 9220 Aalborg, Denmark; vku@bio.aau.dk (V.K.); rw@bio.aau.dk (R.W.)

**Keywords:** mixed PET waste, bioprocess, depolymerization

## Abstract

In recent years, several plastic-degrading enzymes with efficient depolymerization abilities for PET have been reported. Here, we report a bioprocess for mixed PET waste depolymerization using crude extracellularly expressed enzymes in *E. coli*. The enzymes, namely FastPETase, LCC, and LCC^ICCG^, were screened to depolymerize amorphous PET powder and films of different sizes and crystallinity. FastPETase, LCC, and LCC^ICCG^ achieved approximately 25, 34, and 70% depolymerization, respectively, when applied to 13 g L^−1^ of PET film, powder, or mixed waste in optimized enzyme conditions without any pH control. The yield of terephthalic acid in the hydrolytic process was maximum for LCC^ICCG^ followed by LCC and FastPETase. Finally, extracellular LCC^ICCG^-producing *E. coli* cells were cultivated using minimal media supplemented with 0.1% ammonium chloride and 1% glycerol as nitrogen and carbon sources in a bioreactor with a final protein content and specific activity of 119 ± 5 mg L^−1^ and 1232 ± 18 U mg^−1^, respectively. Nearly complete depolymerization of 13 g L^−1^ PET and 23.8 g L^−1^ post-consumer PET was achieved in 50 h using crude LCC^ICCG^ supernatant, without enzyme purification, at 62 °C. A bioprocess was thus developed to depolymerize 100 g L^−1^ mixed PET trays and bottle waste (MW1 and MW2), reaching 78% and 50% yield at 62 °C with a crude enzyme loading of 2.32 mg g^−1^ PET in 60 h. The results demonstrate an easy PET depolymerization strategy that could be exploited in large-scale facilities for efficient plastic waste treatment.

## 1. Introduction

Polyethylene terephthalate (PET) is one of the most important plastics in volume, accounting for 18% of global plastic production. The huge consumption and usage generate a huge amount of discarded PET plastics, and its subsequent mismanagement severely damages the ecological environment [1,2,3,4,5]. Globally, only 9% of plastic waste is recycled, leaving the rest accumulating in landfills or natural environments, where it persists for centuries. Approximately 19–23 million tons of plastic waste leaks into aquatic ecosystems, polluting lakes, rivers, and seas, and this figure could triple by 2040 without intervention (UNEP). The Ellen MacArthur Foundation report (breaking the plastic wave) predicted that there will be more plastic waste than fish biomass in the sea by 2050. A recent study employed machine learning to foresee that annual mismanaged plastic waste will nearly double to 121 million metric tons by 2050. Annual greenhouse gas emissions from plastic waste are also projected to grow by 37% to 3.35 billion tons of CO_2_ [6]. This underscores the urgent need for sustainable waste management solutions. To overcome this problem, PET plastics recycling (thermo-chemical, mechanical, and biological) has been envisaged as an efficient and effective approach. It extends PET’s life cycle, reduces non-renewable resource consumption, decreases plastic pollution, and protects the ecological environment [5,6]. However, PET thermo-mechanical recycling approaches have some drawbacks related to waste sourcing and their decreased mechanical properties during the extrusion process [7]. The chemical hydrolysis of PET (polyethylene terephthalate) involves breaking its ester bonds in the presence of water, to produce terephthalic acid (TPA) and ethylene glycol (EG). This process, often catalyzed by acids such as sulfuric acid or bases, e.g., sodium hydroxide, is used for PET depolymerization and recycling [5,6,7]. However, the chemical hydrolysis of PET requires high temperatures, uses very large amounts of chemicals, is energy-intensive, and can release harmful compounds. The process often generates unwanted side products, leading to purification challenges and potential environmental pollution [6,7]. Mixed plastic waste further complicates these processes due to its complex nature (types, composition, color, and additives). It is difficult to recycle it due to different polymer types and contamination. The recycling process for mixed plastics is energy-intensive, less efficient, and results in downcycled materials with limited utility due to cross-contamination. Biological depolymerization mediated by enzymes has emerged as an efficient and sustainable alternative for plastic treatment. Plastic waste recycling negates harmful environmental consequences by utilizing less energy-intensive processes, and fewer chemicals, and recovering valuable components from plastic waste. The enzymatic hydrolysis of polyethylene terephthalate (PET) offers a sustainable recycling pathway. It aligns with global efforts to reduce plastic pollution and foster a more sustainable materials economy [1,8,9,10]. This approach helps close the loop in a circular economy and offers a more environmentally friendly alternative to chemical recycling methods, which often involve harsher conditions and generate more waste [11]. PET monomers are linked by ester bonds, which can be enzymatically hydrolyzed, producing terephthalic acid (TPA) and ethylene glycol (EG) [2]. Several hydrolases, including lipases, cutinases, and esterases, have been tested for PET hydrolysis. Lipases primarily act on long-chain triglycerides and require hydrophobic interfaces for activation, making them less efficient on PET. Cutinases, such as *Thermobifida fusca* cutinase, show better PET degradation but with a limited substrate binding ability. Esterases generally hydrolyze short-chain esters and lack the structural adaptations needed for PET polymer breakdown. In contrast, PETases, like those from *Ideonella sakaiensis*, have a uniquely open active site and enhanced substrate binding regions [1,12]. After the discovery of PET hydrolyzing cutinases two decades ago, several hydrolases have been identified and improved for PET hydrolysis [1,7,12,13,14]. Recently, two mutants of thermostable *Is*PETase [15,16] from the bacterium *Ideonella sakaiensis*, namely FastPETase [17] and HotPETase [9], were reported for improved PET depolymerization. In another study, a leaf-branch compost cutinase (LCC) [18] was engineered to obtain the LCC^ICCG^ mutant [19]. The enzyme paved the way for the first industrial PET recycling plant unit in France run by Carbios. Recently, Arnal et al. [2] compared four reported PET hydrolases (LCC^ICCG^, FAST-PETase, HotPETase, and PES-H1^L92F/Q94Y^) and demonstrated their ability to perform PET hydrolysis in industrial settings. They also discussed critical parameters to upscale an enzyme process for industrial deployment indicating that the crystallinity of PET, surface area, reaction temperature, the efficiency of the enzyme vs. recrystallization, and substrate loading are among the most critical factors. It is now well known that PET hydrolases prefer amorphous regions of PET and no PET hydrolases have been reported to act efficiently on highly crystalline forms of the polymer, typically found in consumer products [2,20,21]. It is necessary to perform a pretreatment to convert PET to its amorphous state to meet the high enzymatic conversion essential to meet techno-economic goals. The thermostability, expression yields, and enzyme activity are also crucial factors in reaching industrial scales. The pretreatment of postconsumer waste and crystallinity of PET powder and enzyme production platform negatively affect the overall process. The extent of PET breakdown, substrate loading, enzyme price (production and purification cost), and enzyme loading are all key cost drivers [22]. The enzyme purification process requires high energy, chemicals, and water consumption with an overall negative impact on the environment and process economy. Therefore, it would be worthwhile to assess the potential of crude enzyme on PET hydrolysis focusing on high yield enzyme production in the extracellular supernatant. In this study, we have compared the depolymerization efficiency of three PET hydrolases (FastPETase, LCC, and LCC^ICCG^) on different PET substrate types such as powder or films, amorphous or crystalline, pure or mixed waste, and optimized key variables in small-scale. We demonstrated the extracellular expression of LCC^ICCG^ in *E. coli* and the utilization of crude supernatant to catalyze efficient depolymerization of pure PET powder, films, post-consumer trays, and mixed trays and bottle waste (originated from Spain). A bioprocess has been developed for the efficient depolymerization of mixed PET waste in a 1 L bioreactor and subsequent recovery and purification of TPA from the reaction mixture. This work provides an important foundation for a robust one-pot (2-steps) enzymatic production and hydrolysis process that can be extended to high-expression hosts to minimize the cost of enzyme production and purification, thus further improving the PET depolymerization viability. This approach opens for potentially more sustainable and low-cost PET hydrolytic processes to biotechnologically recycle real mixed PET waste.

## 2. Materials and Methods

### 2.1. Materials, Plasmids, and Enzymes

All the chemicals used in this study were of analytical grade. The p-Nitrophenol butyrate (pNPB), TPA, and MHET were purchased from Sigma-Aldrich (St. Louis, MO, USA). All the plasmids and gene constructs were synthesized by Gene Universal (Newark, DE, USA). The sequences for FastPETase, LCC, and LCC^ICCG^ [17,18,19,20] were retrieved from the PDB database, codon-optimized, cloned into pET26b (+) and pET28a (+) vector from Gene Universal (Newark, DE, USA). PET films (0.25 mm amorphous and crystalline) were purchased from Goodfellow Cambridge Limited (Hamburg, Germany). PET powders of different sizes (0.05, 0.1, 0.25, 0.75; Table 1) were kindly provided by Dr. Alessandro Pellis, University of Genova, Italy, and cryo-milled to obtain different sizes and crystallinity. Mixed plastic waste was received from different sources/origins and assigned the acronyms MW1-2 (Table 1) and cryo-milled PET powder (0.08 mm) as part of the UPLIFT project from AIMPLAS, Valencia, Spain. This waste was composed only of PET (MW1 and MW2) and were micronized to obtain the sample size of 1 mm in diameter using an ultra-centrifugal grinder (RetschTM 171 ZM200, ATS Scientific, Madrid, Spain) or cryo-milled to obtain different sizes (Table 1).

### 2.2. Expression and Purification of Enzymes

FastPETase, LCC, and LCC^ICCG^ were expressed in the electrocompetent *Escherichia coli* strain BL21 (DE3) using pET28a (+) (FastPETase) and pET26b (+) vector (LCC and LCC^ICCG^). The sequence of LCC and LCC^ICCG^ contained N-terminal PelB peptide for periplasmic expression. A single colony of *E. coli* BL21 (DE3) cells containing the gene FastPETase or LCC or LCC^ICCG^ was transferred in 5 mL LB medium or M9 medium (Appendix A) supplemented with 50 µg mL^−1^ kanamycin and incubated at 37 °C for 5 h. One mL of starter culture was transferred to a 50 mL LB_Kan_ medium or M9 medium and incubated at 37 °C and 190 rpm till the OD_600_ reached 0.6–0.8. The protein production was induced using 0.5 mM (FastPETase) or 1 mM (LCC and LCC^ICCG^) IPTG and further incubated at 20 °C (FastPETase) and 30 °C (LCC and LCC^ICCG^) for 24 h, respectively. The cells were harvested by centrifugation at 8000× *g* for 20 min at 4 °C. For FastPETase, the cell pellet was resuspended (10% wet weight) in lysis or sample loading buffer (50 mM potassium phosphate buffer, 300 mM NaCl, 5 mM Imidazole; pH 7.5). The cells were lysed using sonication (45% amplitude; 9 s on, 10 s off; 15 min, and 2 cycles). The lysate was centrifuged at 12,000× *g* for 30 min and pellet was discarded and the supernatant was used for further purification. For LCC and LCC^ICCG^, the supernatant was retained, and the pH was adjusted to 7.5. The supernatant was equilibrated with imidazole (5 mM) and 300 mM NaCl to mimic loading buffer conditions. The HisTrap FF column was pre-equilibrated (5 column volume) with 50 mM potassium phosphate buffer, 300 mM NaCl, 5 mM Imidazole; pH 7.5 (AKTA explorer system, GE Healthcare). The 0.22 µm filtered cell-free supernatant was loaded onto the column 2 mL min^−1^). After loading, the column was washed with wash buffer (50 mM potassium phosphate buffer, 300 mM NaCl, 20 mM Imidazole; pH 7.5) with 4 column volumes (2 mL min^−1^) to remove unbound proteins. The protein was eluted (1 mL min^−1^) with elution buffer (50 mM potassium phosphate buffer, 300 mM NaCl, 200 mM Imidazole; pH 7.5). The eluted fractions were checked on the SDS PAGE. The purified fractions were pooled, dialyzed, and concentrated using 3 kDa Amicon filters (4000 g) against 50 mM potassium phosphate buffer (pH 7.5). The purified enzyme was used for the depolymerization of PET powder and film.

### 2.3. Estimation of Esterase Activity and Protein Concentration

Esterase activity was measured at 50 °C (FastPETase) and 65 °C (LCC and LCC^ICCG^) using p-NPB as a substrate as previously reported [23,24]. Solution A was composed of 86 µL of 50 mM p-nitrophenol butyrate (p-NPB) and 1000 µL of 2-butanol and Solution B was prepared using 40 µL of Solution A and 1 mL of 50 mM potassium phosphate buffer (pH 8.0). A final assay mixture made up of 200 µL of Solution B and 20 µL of enzyme solution (diluted if required). The change in absorbance was recorded at 405 nm due to the hydrolytic release of p-nitrophenol (p-NP), which was measured using a microplate reader (Biotek, Santa Clara, CA, USA). One unit is defined as the enzyme required to hydrolyze 1 µmol of substrate per minute under the given assay conditions. The standard of p-NP was prepared (10 –M-100 µM). The protein concentration was determined using Bradford assay [23,25]. Briefly, 4 µL of the protein solution was added into the wells of a 96-well micro-titer plate and 196 µL of Bradford solution and incubated for 5 min. The potassium phosphate buffer (50 mM, pH 8.0) was used as blank, and the absorption was measured after 5 min at 595 nm. The protein concentration was calculated using BSA as a standard (1–100 µg mL^−1^).

### 2.4. Evaluating Enzyme Performance on a Small Scale on Different Powder and Films

For all the experiments, 13.0 ± 0.5 mg of PET amorphous film (size 0.5 × 1 cm, thickness 0.25 mm GoodFellow) or powder (Table 1) were incubated with 0.5 mg g^−1^ PET of purified enzyme (FastPETase, LCC, and LCC LCC^ICCG^) in 1.0 mL buffer (50 mM phosphate buffer, pH 8.0) in separate reactions. The Eppendorf tubes were incubated in a thermo-shaker (constant vertical shaking) at an optimum temperature of 50 °C [13], 65 °C, and 72 °C [19] for FastPETase, LCC, and LCC^ICCG^, respectively. The blank (buffer) and control (film/powder and buffer without enzymes) were also incubated. The reaction was monitored for 96 h and the released monomers were quantified using HPLC-DAD analysis. The monomer concentration is expressed and presented as molar yield (Appendix A).

### 2.5. Evaluation of Different Reaction Variables for PET Hydrolysis

Different 50 mM buffer systems, such as KH_2_PO_4_-K_2_HPO_4_ (pH 6.5–7.5), KH_2_PO_4_-NaOH (pH 8.0–9.0), glycine-NaOH (pH 9.0–10.0), NaHCO_3_-Na_2_CO_3_ (pH 10.0–11.0), were tested for the depolymerization of PET film. Different enzyme concentrations (mg g^−1^ PET) for FastPETase, LCC, and LCC^ICCG^ (0.1, 0.25, 0.5, 1, and 2 mg g^−1^ PET) were used to optimize the effect of enzyme loading on PET depolymerization. 13.0 mg of PET powder (0.08 mm cryo-milled amorphous) and film (0.2 mm amorphous) were incubated in 1 mL KH_2_PO_4_-NaOH buffer (50 mM, pH 8.0) for 96 h in different enzyme concentrations. The incubation temperature (30, 40, 50, 65, 72, and 80 °C), and reaction time (2, 4, 8, 16, 24, 30, 36, 48, 60, 72, and 96 h) were also studied.

### 2.6. Evaluating Enzyme Performance in 10 mL (Without pH Control) and 1 L Bioreactor (pH Control)

The depolymerization of PET film (0.25 mm amorphous), PET powder (0.08 mm amorphous), PET tray waste, and PET bottle waste from Spain was performed in 10 mL falcon tubes in the optimized conditions without any pH control and subsequently in a 1 L bioreactor (Biostat A, Sartorius, Göttingen, Germany) under controlled conditions. 13 g L^−1^ PET film was added to 1000 mL of 50 mM phosphate-NaOH buffer (pH 8.0) and 0.5 mg g^−1^ PET (FastPETse), and 1.0 mg g^−1^ PET (LCC and LCC^ICCG^) in separate reactions. The reaction was performed at 50, 65, and 72 °C for FastPETase, LCC, and LCC^ICCG^, respectively. The initial pH of the reaction was adjusted to 8.0 using 1 M NaOH. The samples were collected at different time intervals and monomers were quantified using HPLC analysis. The reaction was stopped after 96 h, residual PET films were recovered from the reaction, and weight loss (Appendix A) was measured.

### 2.7. Production of LCC^ICCG^ in the Bioreactor and PET Hydrolysis

For seed preparation, *E. coli* containing LCC^ICCG^ was streaked on LB agar plates (50 mg L^−1^ Kanamycin) and incubated at 37 °C overnight. One single colony was inoculated to 5 mL M9 medium (Appendix A) in a 50 mL falcon for overnight growth at 37 °C and 180 rpm. A 1 mL measurement of this seed culture was transferred to a 250 mL flask containing 50 mL fresh M9 medium to grow for another 12 h until an OD_600_ of 1.2–1.8 was reached. This culture was used to inoculate the 2 L batch bioreactor (Biostat A, Sartorius) at 2% (*v*/*v*). The exponentially growing cells from the seed culture were transferred into the bioreactor to initiate the fermentation (*t* = 0 h). The dissolved oxygen level of the growth experiments was set at 20% of air saturation by cascade controls of agitation speed. A two-stage temperature control was implemented, i.e., 37 °C for the growth phase during 0–4 h and 30 °C for enzyme production in the remainder of the run. The pH was maintained at 7.2 throughout the run by feeding 1 M sodium hydroxide solution or 1 M HCl solution. When the O.D_600_ reached 0.6–0.8, the temperature was lowered to 30 °C and the protein production was induced with 1 mM IPTG and kept for growth till 24 h. The cell debris was removed by centrifugation at 10,000× *g* for 15 min. The pH of the supernatant (crude LCC^ICCG^) was adjusted to 8.0 (using 1 M NaOH) in the same reactor, and the reaction temperature was adjusted to 62 °C (the maximum temperature that could be reached in the bioreactor) and at a fixed rpm speed of 600 using a Rushton impeller to provide efficient mixing. Then, 13 g L^−1^ PET material (film or powder) was added to the supernatant and monitored for 96 h. The pH of the reaction was maintained at 8.0 with 1 M NaOH. The monomers and oligomers released were measured using HPLC at different intervals, and quantitative NMR was used to calculate the mass balance.

### 2.8. Depolymerization of Mixed PET Waste Using LCC^ICCG^ in 1 L

LCC^ICCG^ was produced using *E. coli* BL21 cells as described in the above section in a 2 L bioreactor as described earlier. After growth, the cell debris was removed and the supernatant was concentrated using a filtration device (10 kDa, Merck Millipore, Burlington, MA, USA) to 1 L. Different PET materials i.e., PET tray (commercial; 24 g L^−1^), PET tray waste (100 g L^−1^), and PET bottle waste (100 g L^−1^) were depolymerized in separate experiments using approximately 0.8 mg g^−1^ PET of ICCG in a 1 L bioreactor (Biostat A- Sartorius) under cascade pH control (pH 8.0 using 1 M NaOH) and 62 °C for 96 h. The temperature regulation was performed in the water-jacketed bioreactor, and a Rushton impeller was used to maintain constant agitation at 600 rpm. The reaction was initiated by the addition of PET substrate (mix tray or bottle waste) into the reactor containing crude LCC^ICCG^ at 62 °C, 600 rpm, and cascade pH control at 8.0. The samples of 5 mL from the PET hydrolysis were collected at different time intervals from the sampling port. After the completion of the reaction, the residual PET and protein were removed by centrifugation. TPA was recovered from the reaction mixture using a modified protocol [26]. The purification of TPA from PET hydrolysate was performed by increasing the pH to over 9.0 to ensure the solubilization of all the monomers followed by centrifugation at 10,000 rpm to remove any debris of denatured proteins. The pH of the supernatant was decreased (to 2.0) stepwise to monitor the precipitation of TPA. The precipitates were washed with cold acidified water, lyophilized, and analyzed by ^1^H-NMR for purity.

### 2.9. Advanced Analysis

#### 2.9.1. High-Performance Liquid Chromatography (HPLC)

HPLC analysis was performed for all the PET hydrolysis reactions to quantify the monomers (TPA, MHET released). Next, 500 μL samples were taken, centrifuged at 20,000 rpm for 10 min, and the supernatant was diluted 1:1 with methanol. TPA, MHET, and BHET standards were prepared in 50% methanol (*v*/*v*). The samples were analyzed using a Dionex Ultimate 3000 system, fitted with a diode array detector. A C18 column (Phenomenex Luna 5 μm, 250 mm × 4.6 mm) at 30 °C was used for the separation of the products of the reaction, with 0.1% formic acid in water (A) and methanol (B) as the mobile phase. The solvent gradient was as follows: 10% B (0–5 min), increased linearly to 50% (5–17 min), and 100% B (17–20 min). The flow rate was 1 mL min^−1^ and the injection volume was 2 μL, with detection at 241 nm and a total run time of 20 min.

#### 2.9.2. Nuclear Magnetic Resonance (NMR)

Samples were prepared by mixing 500 μL of supernatant from the enzymatic reaction with 25 μL of a 200 mM solution of 3-(Trimethylsilyl)-2,2,3,3-tetradeuteropropionic acid sodium salt (TSP-D4) in D_2_O. The methyl signal of TSP served as an internal standard for the chemical shift (δ = 0 ppm). ^1^H-NMR spectra were recorded on a BRUKER AVIII-600 MHz NMR spectrometer, equipped with a cryogenic CPP-TCI probe (Bruker Scientific LLC, Billerica, MA, USA). Spectra were recorded at 298.1 K with a standard 1D pulse sequence with an acquisition time of 2.73 s (64 k complex data points, spectral width of 20 ppm). The relaxation delay was set to 25.5 s. During the last 5 s of the relaxation delay, a weak continuous-wave irradiation of γB_1_/2π = 70 Hz was applied for water suppression. Resonance assignment in product mixtures was conducted by comparing chemical shifts with spectra of pure standards [11].

#### 2.9.3. Differential Scanning Calorimetry (DSC) Analysis of PET Materials

The thermal behavior of the PET film and powder was determined by using DSC. The DSC experiment was carried out using NETZSCH Proteus^®^ for Differential Scanning Calorimetry (DSC). In the first scan, each sample was heated to 300 °C at a heating rate of 15 °C min^−1^. The sample was then cooled at the same rate before reheating the samples to 300 °C. The test samples, weighing about 8.0 ± 2.0 mg, were placed in a T-zero thermionic aluminum pan for analysis and all tests were carried out under a nitrogen atmosphere. The percentage of crystallinity (Xc) was calculated based on the equation (Equation (1)) below:(1)$$Xc=\frac{\Delta Hm-\Delta Hc}{wt*\Delta Hf}\times 100$$

∆H_m_ is the enthalpy of melting that can be determined by integrating the endothermic melting peak; ∆Hc is the enthalpy of cold crystallization and determined by integrating the exothermic cold crystallization peak; wt is the weight fraction of polyester in the plastic; and ∆H_f_ 100% is the enthalpy of melting for a fully crystalline polymer and taken from the literature as 140.1 J g^−1^.

## 3. Results and Discussion

### 3.1. Expression, Production, and Purification of FastPETase, LCC, and LCC^ICCG^

FastPETase, LCC, and LCC^ICCG^ were expressed in the *Escherichia coli* strain BL21 (DE3). FastPETase was obtained from the cytoplasmic fraction (Figure 1a) after the cell lysis whereas LCC and LCC^ICCG^ were obtained in extracellular supernatant (Figure 1b,c and Appendix A). LCC and LCC^ICCG^ contain N-terminal pelB leader peptide that guides protein into the periplasmic space. The pelB peptide does not correspond to the active transport of the enzymes to extracellular space. However, the enzymes were recovered from the culture supernatant. It could be attributed to the phospholipase activity of these enzymes especially cutinases [27,28,29] which makes the cell membrane permeable and the cause leaking of the enzyme out of the cell. There was no cell cytotoxicity observed during the growth phase and enzyme production. After expression and purification, a final yield of 26, 35, and 32 mg L^−1^ of culture was obtained with a specific activity of 968 ± 15, 1430 ± 27, and 2187 ± 19 U mg^−1^ for FastPETase, LCC, and LCC^ICCG^, respectively (Appendix A).

### 3.2. Enzymatic Depolymerization of Different Powders and Films in Small-Scale

FastPETase, LCC, and LCC^ICCG^ are among the best polyester hydrolases reported so far and are known to depolymerize aliphatic and aromatic polyesters including PET [2,17,18,19]. The purified enzymes were used for PET depolymerization experiments on a small scale (1 mL) without any pH control. Different PET powders and films have been used in this study and all the materials were characterized using DSC analysis (Table 1 and Appendix A, Appendix A). Preliminary tests were made, using the purified enzyme (0.5 mg g^−1^ PET), 13 g L^−1^ of PET film or powder, and 50 mM bicin NaOH buffer at the respective optimum temperature of the enzyme, i.e., 50, 65, and 72 °C for FastPETase, LCC and LCC^ICCG^ in 1.0 mL reactions. The specific activities of the enzymes were calculated using the pNPB assay and expressed as U/mg and an equal number of units were added to the reaction mixture in all depolymerization assays. Then, 968 ± 15, 1430 ± 27, and 2187 ± 19 U mg^−1^ were used for FastPETase, LCC, and LCC^ICCG^, respectively. FastPETase had lower molar yields (%), i.e., 6.2, 12.3 (P80A), and 6.0 and 10.8% (P750A) of TPA and MHET, respectively, relative to the total PET compared to the other two enzymes (Figure 2a). LCC treated P80A and P750A reactions molar yield of 12 and 18.1% TPA and 16.7 and 15.5% MHET, respectively, (Figure 2b). LCC^ICCG^ exhibited a maximum yield of TPA (57 and 58 mol%) and MHET (15.3 and 16 mol%) during the depolymerization of cryo-milled PET powders (P80A, P750A) (Figure 2c). The depolymerization of PET 0.2 mm Goodfellow PET by FastPETase resulted in 20% weight loss (Figure 2d) which was lower than LCC^ICCG^ (60%) and LCC (27%). LCC^ICCG^ treated PET film (F250A) also had maximum weight loss (60%) and yielded 45.1% TPA and 12.3% of MHET (Figure 2f). LCC^ICCG^ seems to have better MHET hydrolysis activity, better substrate binding, and optimized enzyme thermal stability under the tested buffer conditions that contributed to the increased TPA yield compared to LCC and FastPETase.

The idea was to use powder of different sizes and crystallinities to see its effect on the enzymatic reaction. It is anticipated that smaller powder sizes increases surface area and reduces diffusion limitations, thereby improving enzymatic hydrolysis. The effect of increasing crystallinity of powder and films on the depolymerization and monomer release by these enzymes was evident where the enzyme efficiency was severely reduced both for powders and films (Figure 2a–f). LCC^ICCG^ was the most active enzyme on crystalline PET (Figure 2f) but had three times less depolymerization than on amorphous PET. The crystallinity, chain mobility, molecular size, surface topography, and hydrophobicity of PET greatly influence enzyme performance [11,29,30,31,32]. A crystallinity exceeding 20% has been proposed to significantly impede the enzymatic depolymerization process as evidenced in quenched PET powders (27.6% crystallinity) [2,12]. Additionally, the crystalline PET powder of different sizes did not show any improvement in terms of total depolymerization by the enzymes (Figure 2a–c). Brizendine et al. [33] also demonstrated that particle size reduction in PET only affects the initial rate of hydrolysis but not the overall conversion. PET hydrolases adsorb non-specifically on the PET surface, resulting in either a productive complex leading to PET degradation by binding close to the active site, or in an unproductive complex [31,34]. According to the Sabatier principle, when interactions between enzyme and PET (substrate) are of intermediary strength, the reaction is optimal [35,36].

### 3.3. Different Variables Influencing Enzymatic PET Hydrolysis

After the preliminary tests, P80A and F250A were used in subsequent experiments. Enzyme loading for efficient conversion is an important parameter of enzymatic processes such as PET hydrolysis. PET weight loss and monomer release reached a maximum (24.4% weight loss, molar yield of 10.3, and 15% of TPA and MHET) when 0.5 mg g^−1^ PET FastPETase was used at 50 °C in potassium phosphate-NaOH buffer (pH 8.0) after 96 h (Figure 3a). Interestingly, the PET hydrolysis did not improve when the concentration of FastPETase was increased to 1 and 2 mg g^−1^ PET. A recent study showed the low thermostability of FastPETase during PET hydrolysis at different reactor scales compared to other variants [2]. LCC on the other hand showed increased depolymerization of both powder and film (Figure 3b) with an increase in enzyme loading, reaching a maximum weight loss of 32% at a concentration of 1 mg g^−1^ PET. The performance of ICCG at 1 mg g^−1^ PET enzyme loading was superior where the depolymerization was approximately 58% and TPA and MHET molar yield of 55% and 7%, respectively. LCC^ICCG^ has been reported to convert 200 g L^−1^ amorphized PET to 90% over 10 h at 3 mg g^−1^ PET enzyme loading [19]. Similar results were obtained by Brizendine et al. [33] when depolymerization was performed at 65 °C and 100 g L^−1^ loading of PET film (1 × 1 cm) but in 48 h. The amount of TPA compared to MHET also kept increasing (Figure 3c) with increased enzyme loading. A possible explanation could be that the enzyme performs endo cleavage first on the polymer, so at lower enzyme concentrations, more oligomers are released than monomers or MHET is primarily released but then hydrolyzed further.

The effect of using different buffers and pH for PET hydrolysis was investigated. FastPETase (50 °C), LCC (65 °C), and LCC^ICCG^ (72 °C) treated PET exhibited weight loss in all buffer solutions with a pH ranging from 6.5 to 11.0 (Appendix A). The weight loss was maximum in 50 mM potassium phosphate NaOH buffer (at pH 8.0) for all three enzymes and efficient depolymerization in the pH range of 7.0–9.0. Also, PET powder depolymerization was most efficient in 50 mM KH_2_PO_4_-NaOH buffer (pH 8.0 and 9.0) for FastPETase (22%), LCC (31%) and LCC^ICCG^ (58%). There was a rapid decline in depolymerization efficiency as the pH dropped below 7.0 due to the release of acidic hydrolysis products (TPA and MHET) which decreased the reaction mixture’s pH.

Interestingly, LCC^ICCG^ catalyzed PET hydrolysis produced more TPA than MHET compared to LCC and FastPETase in the buffer conditions tested. Here, the lower molarity (50 mM) was chosen to mimic the industrial reaction environments where hydrolysis is performed in water rather than specialized buffered solutions to reduce operational costs [2].

The temperature of incubation was also optimized for these enzymes (Appendix A). All these enzymes exhibited maximum depolymerization at their respective reported optimum temperature, i.e., 50 °C, 65 °C, and 72 °C for FastPETase, LCC, and LCC^ICCG^, respectively [17,18,19] (Appendix A). As expected, the thermostability analysis suggested that FastPETase is least stable in all tested temperatures (50, 60, and 72 °C) after 36 h followed by LCC and LCC^ICCG^ (Appendix A). Furthermore, PET hydrolysis efficiency was monitored over time. It can be seen from Figure 3d that FastPETase achieved maximum depolymerization and molar yield (13.75% MHET and 8% TPA) in 33 h. The depolymerization process slowed down after 36 h and no significant changes were observed after longer incubation times (Figure 2c). This can be due to poor thermostability and short half-life of FastPETase, changes in pH or a combination thereof [2]. LCC showed an improved depolymerization performance (Figure 3e). The released monomer (for both powder and film) increased with prolonged incubation time and molar yield of 26% and 24% of TPA and 14.5% and 13% of MHET (P80A and F250A) in 60 h. LCC^ICCG^ also exhibited maximum depolymerization after 60 h (Figure 3f) but with a higher molar yield of TPA (54%) and less MHET (7%). The TPA yield marginally increased to 56.5% after 72 h and remained constant after that. Brizendine et al. [33] observed differences between the large-scale (bioreactor conditions) and small-scale reactions for MHET concentrations where large-scale reactions generally built up less MHET than small-scale ones. LCC^ICCG^ will preferentially release TPA due to the inhibitory effect of MHET. The reactions can be complemented by adding a carboxylesterase or MHETase to further improve the depolymerization efficiency [37,38].

### 3.4. PET Depolymerization in 10 mL

The previous experiments were performed on a 1 mL scale (50 mM phosphate-NaOH, pH 8.0) without a constant pH control, which may have a significant effect on the overall depolymerization of PET [2]. The first scale-up (10 mL) was performed in the reaction conditions optimized earlier with P80A, F250A, and mixed PET waste MW1, and MW2 without any pH control. The overall depolymerization improved compared to the 1 mL scale. The depolymerization of P80A, F250A, and mixed PET waste MW1 (X_C_ 16%), and MW2 (Xc 28%) by FastPETase led to a weight loss of 25, 24, 23, and 15%, respectively (Figure 4a). LCC catalyzed depolymerization of PET reached 34% weight loss for F250A and MW1, but only 22% weight loss for MW2. ICCG outperforms the other two in terms of weight loss and monomer molar yield for all the substrates tested (Figure 4a). A weight loss of 69, 68, 63, and 56% was recorded for the depolymerization of P80A, F250A, MW1, and MW2, respectively. The release of TPA was maximum for LCC^ICCG^ followed by LCC and FastPETase (Figure 2b). The interesting result is the depolymerization of MW2 (X_c_ = 28%) by ICCG where 57% weight loss (molar yield of 43% TPA and 11% MHET) was achieved compared to LCC (22% weight loss, 8.9% and 11.6% molar yield of TPA and MHET). There is a difference in the observed weight loss of powders and the mass loss obtained from molar yield due to the loss of powder particles during washing, processing, and analysis. Thomsen et al. [31] demonstrated that LCC^ICCG^ performs relatively better than PETase at higher crystallinity (15–25%); however, the depolymerization efficiency was decreased as the crystallinity was increased.

### 3.5. Depolymerization of Different PET Substrates Using Crude LCC^ICCG^ in a Bioreactor (1 L)

In the PET hydrolytic process, pH is a crucial factor with a huge impact on the overall depolymerization. It significantly affects enzyme stability, activity, substrate binding, and product solubility. Consequently, the efficiency of PET hydrolases depends on the optimal pH conditions that support catalytic activity and structural integrity. We performed depolymerization of different PET substrates (PET film, Powder) in the 1 L reactor in the optimized conditions, i.e., 13 g L^−1^ substrate loading (PET film or powder). The *E. coli* BL21 cells containing LCC^ICCG^ were grown in an M9 medium (0.1% ammonium chloride and 0.5% glycerol, 20% pO_2_, pH 7.2). After induction with IPTG, the cells were grown for 32 h (Figure 5a). The final OD_600_ of 5.7, total protein content in the supernatant of 119 mg L^−1^, and pNPB activity of 1232 U mg^−1^ were reached in the bioreactor. Similarly, a fed-batch fermentation (1 L) using LB medium or autoinduction medium with glucose feeding and IPTG induction led to the production of 12 g L^−1^ total protein (10–15% of the target protein LCC) [39] which is higher than the current study. However, it is to be noted that in the current study, the production of the enzyme was performed in a minimal medium and 0.2% glycerol in a batch reactor. 30% of the total protein produced extracellularly accounts for LCC^ICCG^. The enzyme cost during the PET hydrolysis is a key cost driver and a balanced production process with lower costs and higher enzyme yields will be an advantage [22]. After removing cell debris, one-half of the supernatant (1 L) was used for purification and the other half for the bioprocess using the crude enzyme. With the purified enzyme-catalyzed reaction with film F250A, MHET concentration reached a maximum molar yield of 13.1% after 19 h and decreased after that, whereas TPA formation reached the maximum of 94% molar yield in 72 h (Figure 5b) and was also verified by quantitative NMR (Appendix A). However, almost 90% of TPA was released in 25 h showing the efficiency of this enzyme. The molar yield of TPA (Figure 5c and Appendix A) and powder using the crude supernatant reached 90% in approximately 46 h. However, MHET release was very low compared to TPA. NMR spectrum revealed the presence of isophthalic acid and some unknown aromatic compounds (Appendix A). Another interesting difference is the lag phase during the initial hours (5–7 h) for PET film (using a purified enzyme and crude supernatant) which was not evident for powder. Brizendine et al. [33] also observed a lag phase 1 × 1 cm film which seemed to be dependent on the surface area and enzyme concentration. It can be explained by the fact that the enzyme performs an endo cleavage on PET which would not release monomers during the initial phase [31,33]. They also reported that PET particle size does not have an appreciable effect on total depolymerization [33,40,41].

### 3.6. Mixed PET Waste Depolymerization

The depolymerization efficiency using crude enzyme was further tested on a post-consumer PET tray from the supermarket (used for packaging). The PET tray weighing 23.8 g was cut into pieces (Figure 6), mixed with ICCG crude supernatant (2.3 mg g^−1^ PET), and incubated at 62 °C. A 94% depolymerization (based on residual films) of PET was achieved in 60 h with a TPA molar yield of 97% as measured by HPLC (Figure 6a) and 17.2 g TPA (90% molar yield) was recovered (Figure 6b,c). NMR analysis of the purified and redissolved TPA in water at pH 7.85 shows traces of isophthalic acid and some impurities of glycerol and ethylene glycol. The assignment of NMR resonances is given in Appendix A. Tournier et al. [19] reported that LCC^ICCG^ (3 mg g^−1^ PET) can depolymerize 90% of 200 g L^−1^ PET (amorphized and micronized) in 10 h at 72 °C and a similar conversion was reached at 100 g L^−1^ PET (1 × 1 cm) and 65 °C but in 48 h [33].

In the present study, 94% conversion was reached in 60 h at 2.4 g L^−1^ PET loading with the crude supernatant but at a slightly lower temperature, i.e., 62 °C. Furthermore, we tested the crude supernatant on MW1 (mixed PET waste tray from Spain) and MW2 (mixed PET waste bottles from Spain) at 100 g L^−1^ PET loadings and 2.5 mg g^−1^ PET of crude enzyme (Figure 7a and Appendix A). It is interesting to observe that the depolymerization of MW1 (X_c_ = 16%) was nearly 78% in 60 h (Figure 7b) compared to MW2 (X_c_ = 28%) where 50% depolymerization was recorded. The effect of increasing the crystallinity was evident in the overall depolymerization as previously reported [30,32].

Currently, LCC^ICCG^ is among the best enzymes that depolymerize crystalline PET (Xc < 20%) as efficiently as amorphous PET. However, the depolymerization activity decreases sharply after the crystallinity is increased as observed in this work and reported by others [31,42]. The quantitative NMR analysis of the MW1 and MW2 hydrolysate after 60 h revealed the presence of isophthalic acid, an unknown aromatic compound, and oligomers of MHET and BHET, besides TPA and EG (Appendix A). The spectrum showed predominant signals corresponding to TPA (~8.0 ppm) and EG (4.2 ppm), indicating near-complete PET depolymerization. It reveals multiple intermediate signals, including MHET (8.2 ppm, 2H and ~4.3 ppm, 2H), isophthalic acid (~8.3–8.4 ppm, 1H, 2H), diethylene glycol (~3.9 ppm, 4H), and unknown compounds (~7.8 ppm, 2H; ~3.8 ppm, 2H). Overall, the data demonstrate successful PET breakdown, with varying degrees of intermediates’ accumulation. (Appendix A). TPA tends not to be very soluble in the reaction mixture and will certainly affect the mixing and mass transfer. However, it has been reported that the addition of NaOH neutralizes the acidic products released during PET depolymerization. The base addition also leads to the formation of soluble disodium terephthalate (13% *w*/*w*) between 25 and 70 °C [2,43]. The current process runs at 100 g L^−1^ substrate loading (10%), so the process is well below these limits. If the process is run at 20–30% solid loadings, the problem of solid mixing and mass transfer will arise due to insoluble TPA. However, it can be alleviated using higher concentrations of NaOH (to form TPA salt) and ensuring that TPA salt concentrations do not go beyond the solubility limits. TPA, being an acidic compound, tends to precipitate out of the solution at the pH and temperature conditions typically used during the reaction, especially when the substrate loading is high. While NaOH addition is crucial for neutralizing TPA and enhancing its solubility, it must be carefully controlled to avoid excess, which could lead to salt precipitation or undesired changes in reaction conditions. This balance ensures that TPA remains soluble, enabling efficient PET depolymerization even at higher solid loadings. Recently, Chen et al. [44] expressed and secreted FastPETase in *Pichia pastoris* and used crude supernatant to depolymerize 90% of 5 g L^−1^ PET (0.5%) in 18 h in a 10 L reactor. Similarly, LCC variants were produced in *E. coli,* and crude supernatant was applied to degrade 10 g L^−1^ PET with 97.5% depolymerization [45]. Appendix A summarizes recent studies on PET wastes at the bioreactor scale. However, in the current work, 90% depolymerization of 100 g L^−1^ PET (10%) mixed PET tray waste was achieved in a 1 L reactor. This approach is particularly interesting for large-scale industrial applications, reducing enzyme production costs while maintaining high degradation efficiency, thus making enzymatic PET recycling more feasible and sustainable. However, microbial strains with high expression and secretion ability would be highly beneficial for such processes to be successful. Despite the high depolymerization efficiency of LCC^ICCG^, substrate crystallinity above 25% remains a significant limitation, slowing down the reaction and reducing overall PET breakdown. Additionally, scaling up enzyme production and purification remains a challenge, requiring optimization in high-expression hosts to lower costs and improve industrial feasibility.

## 4. Conclusions

We evaluated the depolymerization efficiency of three efficient PET degrading enzymes namely FastPETase, LCC, and LCC^ICCG^ on different powder and films in several conditions (enzyme loading, buffer, temperature, and time) on a small scale and concluded that LCC^ICCG^ is the best-performing enzyme on amorphous and crystalline PET, and mixed PET waste (tray and bottle waste). As reported earlier, the particle size does not affect the overall depolymerization process. However, if crystallinity of the substrate exceeds 25%, it becomes a bottleneck for efficient depolymerization. LCC^ICCG^ production was scaled to 2 L and a PET depolymerization of >90% was achieved in 50 h at 1.3% and 2.4% loading of amorphous PET and postconsumer PET, respectively. Furthermore, 78% of 100 gL^−1^ MW1 (X_c_ = 16%) and 50% of MW2 (X_c_ = 28%) were depolymerized in 60 h showing the effect of an increase in crystallinity on the depolymerization rate. Our study confirms that LCC^ICCG^ outperforms FastPETase and LCC in PET depolymerization across various conditions, achieving >90% degradation efficiency in small-scale and liter-scale experiments. These findings provide a strong foundation for more robust and viable biotechnological depolymerization of real PET waste, by developing a one-pot enzymatic production and hydrolysis process, reducing purification costs, and thus laying the basis for enhanced industrial PET recycling efficiency.

## Figures and Tables

**Figure 1 polymers-17-00763-f001:**
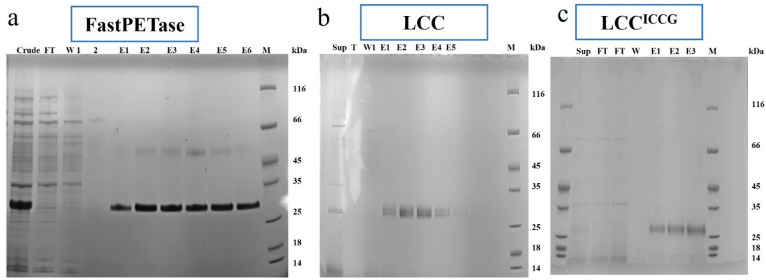
SDS PAGE analysis of (**a**) FastPETase, (**b**) LCC, and (**c**) LCC^ICCG^. Lanes contain different samples and are named as follows: cell lysate (crude), extracellular supernatant (sup), flowthrough (FT) wash (W), eluates (E), and protein marker (M). The proteins appear as a single band (FastPETase 30.1 kDa) with actual molecular weight of 27.9 kDa, (LCC 29.9 kDa) with actual molecular weight of 29.0 kDa, and (LCC^ICCG^ 27.3 kDa) with actual molecular weight of 28.6 kDa. The gels were prepared twice with the gel shown representative of the set.

**Figure 2 polymers-17-00763-f002:**
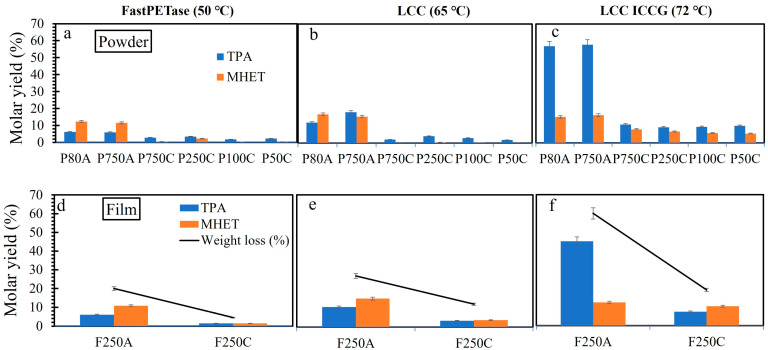
Depolymerization of different PET powders and films (amorphous and crystalline) by (**a**,**d**) FastPETase, (**b**,**e**) LCC, and (**c**,**f**) LCC^ICCG^. The reaction mixture was composed of 50 mM Bicin-NaOH buffer (pH 8.0), 13 g L^−1^ PET powder or film, and 0.5 mg g^−1^ PET enzyme. The reaction was incubated for 96 h at the respective optimum temperature of the enzymes, i.e., 50 °C (FastPETase), 65 °C (LCC), and LCC^ICCG^ (72 °C) without any pH control.

**Figure 3 polymers-17-00763-f003:**
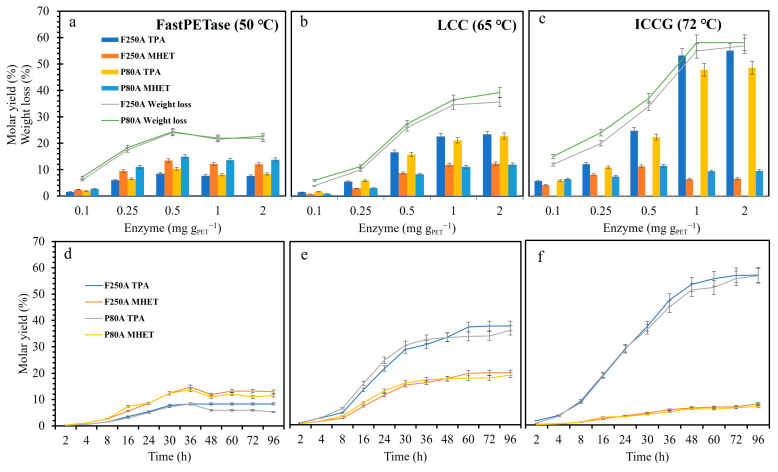
Depolymerization (weight loss and monomer release) of PET powder (P80A) and film (F250A) by (**a**) FastPETase, (**b**) LCC, and (**c**) LCC^ICCG^. The reaction mixture contained 50 mM KH_2_PO_4_-NaOH buffer (pH 8.0), 13 g L^−1^ PET powder, or film at different enzyme loading (0.1, 0.25, 0.5, 1.0, and 2.0 mg g^−1^ PET). It was incubated for 96 h at the respective optimum temperature of the enzymes, i.e., 50 °C (FastPETase), 65 °C (LCC), and LCC^ICCG^ (72 °C) without any pH control. Depolymerization (monomer release) of PET powder (P80A) and film (F250A) by (**d**) FastPETase (0.5 mg g^−1^ PET), (**e**) LCC (1 mg g^−1^ PET), and (**f**) LCC^ICCG^ (1 mg g^−1^ PET) at different time intervals (2, 4, 8, 16, 24, 30, 36, 48, 60, 72, and 96 h) in 50 mM KH_2_PO_4_-NaOH buffer (pH 8.0), 13 g L^−1^ PET powder, or film.

**Figure 4 polymers-17-00763-f004:**
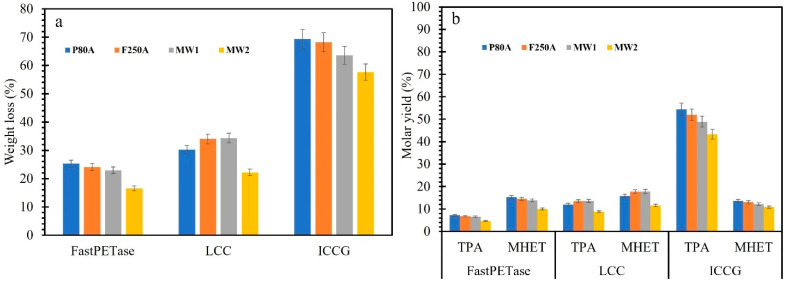
(**a**) Depolymerization (weight loss) of PET powder (P80A) and film (F250A), mixed PET waste MW1, and MW2 by FastPETase, LCC, and LCC^ICCG^. The reaction was performed in a 10 mL volume containing 50 mM KH_2_PO_4_-NaOH buffer (pH 8.0), 13 g L^−1^ PET powder, or film or MW 1 mg g^−1^ enzyme for 48 h at the respective optimum temperature of the enzymes, i.e., 50 °C (FastPETase), 65 °C (LCC), and LCC^ICCG^ (72 °C) without any pH control. (**b**) The released monomers were measured by HPLC after the completion of the reaction and expressed as molar yield (%).

**Figure 5 polymers-17-00763-f005:**
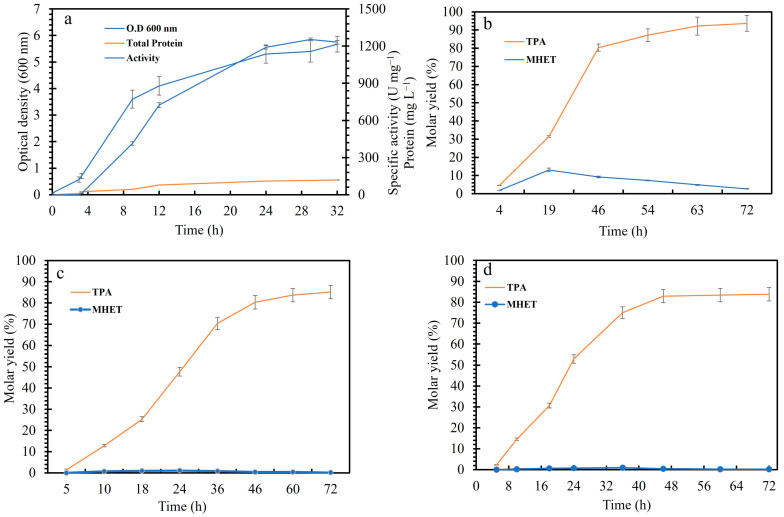
(**a**) Growth profile, total protein content in the supernatant, and specific activity of LCC^ICCG^-producing *E. coli* BL21 cells in a 2 L fermenter after induction with IPTG at 2.2 h (O.D_600_ 0.6–0.8). (**b**) Depolymerization profile (monomer release) of PET film (F250A) using the purified LCC^ICCG^, (**c**) depolymerization of PET film (F250A), and (**d**) PET powder (P80A) using crude supernatant of LCC^ICCG^. The reaction was performed in a 1L volume containing 50 mM KH_2_PO_4_-NaOH buffer (pH 8.0), 13 g L^−1^ PET powder, or film, and 1 mg g^−1^ PET enzyme at 62 °C with pH control (1M NaOH). The monomer release is measured by HPLC.

**Figure 6 polymers-17-00763-f006:**
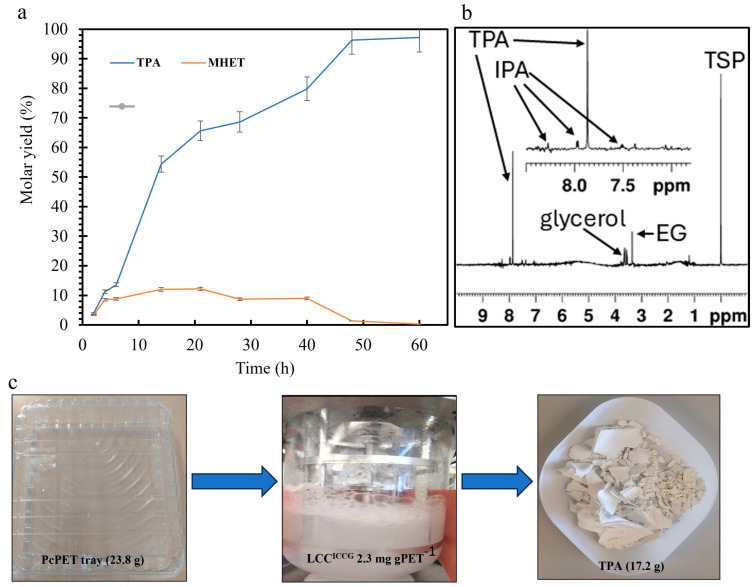
(**a**) Depolymerization profile (mol% of TPA and MHET relative to PET) of pcPET tray (1 × 1 cm) using the LCC^ICCG^. The reaction was performed in a 1 L volume containing 50 mM KH_2_PO_4_-NaOH buffer (pH 8.0), 2.3 mg g^−1^ PET of crude LCC^ICCG^, 23.8 g L^−1^ PET tray at 62 °C with pH control (1 M NaOH). The monomers released were measured using HPLC. (**b**) NMR spectrum of TPA recovered after 60 h. Apart from TPA, the sample contains IPA, which is part of commercial PET, EG from the hydrolysis of PET, glycerol from the culture medium, and TSP-d4 as a chemical shift standard. (**c**) Post-consumer PET tray was used for the enzymatic reaction and TPA recovered after the hydrolysis.

**Figure 7 polymers-17-00763-f007:**
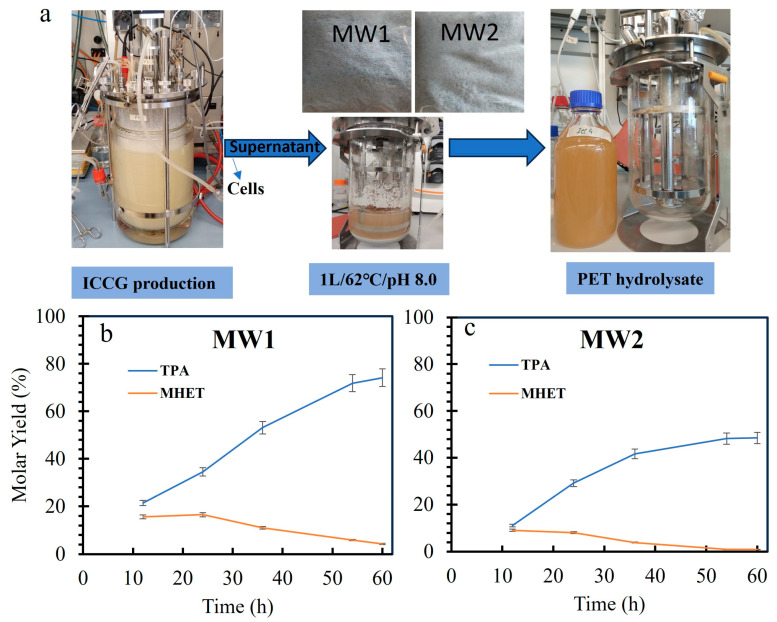
(**a**) Schematic representation of LCC^ICCG^ production using recombinant *E. coli* BL21 cells in the 2 L reactor. The supernatant was concentrated 2 times using ultrafiltration (10 kDa). The mixed plastic wastes MW1 and MW2 were added to the supernatant in separate reactions (100 g L^−1^ PET). The reaction was performed in 1 L volume containing approximately 2.5 mg g^−1^ PET of crude LCC^ICCG^ at 62 °C with pH control (1 M NaOH). HPLC measurements of TPA and MHET during depolymerization of mixed PET waste (**b**) MW1, and (**c**) MW2.

**Table 1 polymers-17-00763-t001:** Size and crystallinity of different PET materials used in this study.

Sample Code	Sample Name	Source	Size or Thickness (mm)	X_c_ (%)
P80A	PET powder cryo-milled	Goodfellow	0.08	9.3
P750A	PET powder cryo-milled	Goodfellow	0.75	16.6
P750C	PET powder cryo-milled	Goodfellow	0.75	37.7
P250C	PET powder cryo-milled	Goodfellow	0.25	42.6
P100C	PET powder cryo-milled	Goodfellow	0.1	39.1
P750C	PET powder cryo-milled	Goodfellow	0.05	37.7
F250A	PET film	Goodfellow	0.25	0.6
F250C	PET film	Goodfellow	0.25	35.1
MW1	PET trays (real waste, origin Spain) cryo-milled at <500 microns	Waste stream	<0.5	15.9
MW2	PET Bottle (real waste, origin: Spain) cryo-milled at >500 microns	Waste stream	<0.5	28

P (powder), A (amorphous), C (crystalline).

## Data Availability

Data will be made available on request.

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
