# Peer review of "Efficient Bioprocess for Mixed PET Waste Depolymerization Using Crude Cutinase"

_polymers, 2025, doi:10.3390/polym17060763_

Round 1

Reviewer 1 Report

Comments and Suggestions for Authors

This manuscript develops a bioprocess for the depolymerization of mixed PET waste using crude enzymes expressed extracellularly in E. coli. The enzymes FastPETase, LCC, and LCCICCG were selected to depolymerize amorphous PET powder and films of different sizes and crystallinity. After investigating the best conditions for depolymerizing crystalline PET and PET waste samples, it was observed that the yield of terephthalic acid in the hydrolytic process was maximum for LCCICCG, followed by LCC and FastPETase. Finally, extracellular E. coli cells producing LCCICCG were grown using a minimal medium supplemented with 0.1% ammonium chloride and 1% glycerol as nitrogen and carbon sources in a bioreactor with a final protein content and specific activity of 119±5 mg L-1 and 1232±18 U mg-1, respectively. Almost complete depolymerization of 13 g L-1 PET and 23.8 g L-1 post-consumer PET was achieved in 50 h using the crude LCCICCG supernatant, without enzymatic purification, at 62 ℃. A bioprocess was thus developed to depolymerize 100 g L-1 of mixed PET trays and bottle waste (MW1 and MW2), achieving 78% and 50% yield at 62 ℃ with a crude enzyme load of 2.32 mg g-1 PET in 60 h.

The work is indeed scientifically rigorous and of great importance in the field, especially with regard to obtaining more sustainable processes and the use of waste such as PET. However, some suggestions and corrections need to be made:

- Introduction section
The authors need to talk about chemical methods of PET hydrolysis and their disadvantages, as well as attempts to hydrolyze PET with other hydrolases, such as lipases, cutinases and esterases, listing their disadvantages compared to PETases;

- Methodology

Table 1: Can the authors provide any explanation for the choice of PET powder particle sizes applied in this work?

Results and Discussion
Item 3.1 - line 278: the authors state that After expression and purification, a final yield of 278 26, 35, and 32 mg L-1 of culture was obtained. However, the authors do not express the specific activities of each enzyme. Provide.

Item 3.2 - line 293: “Preliminary tests were done, using the purified enzyme (0.5 293 mg g-1PET), 13 g L-1 of PET film or powder, and 50 mM bicin NaOH buffer at the respective 294 optimum temperature of the enzyme i.e. 50, 65, and 72℃ for FastPETase, LCC and LCCICCG 295 in 1.0 mL reactions.” The authors describe that they carried out initial PET degradation tests with these enzymes and that the optimum temperature for each was found. What was the main response variable in arriving at these temperatures? 

line 313: The authors start a discussion about the influence of PET crystallinity and enzyme access on polymer degradation, as a way of explaining the differences found in Figure 2. Can't enzymatic activity also have an influence? It is important to note that the authors need to demonstrate the quantities of enzymes used in specific activity, since the mass is conserved in the case of denaturation, but the activity decreases. 

Item 3.3 - Optimization of different parameters for enzymatic PET hydrolysis (It's listed as 2.3, but it should be 3.3, as well as the next item - check): the authors can't say that the system was optimized, because they didn't plan the experiments, but rather checked the variables influencing the system.

- Figure 3 has very low resolution and it is difficult to see the error bars.

- Line 361: Also PET powder depolymeri-zation was most efficient in 50 mM KH2PO4-NaOH buffer (pH 8.0 and 9.0) for FastPETase (22%), LCC (31%) and LCCICCG (58%). There was a rapid decline in depolymerization efficiency as the pH dropped below 7.0 due to the release of acidic hydrolysis products (TPA 363 and MHET) which decreased the reaction mixture's pH. 
Did the authors verify the optimum pH for enzyme activity? This information is important for maintaining catalytic efficiency in the system. I reiterate that the authors need to express the enzyme quantity data in specific activity in the manuscript.

- Line 365: Interestingly, LCCICCG catalyzed PET hydrolysis producing more TPA than MHET compared to LCC and FastPETase in the buffer conditions tested.
To what do the authors attribute this ability?

Line 421: In the PET hydrolytic process, pH is a crucial factor with a huge impact on the overall 421 depolymerization.
Please provide further explanations.

- Figure 7: were no replicates made? Provide erros bars

Author Response

Action Taken Report

Submission ID: polymers-3496310

Title: Title: Efficient bioprocess for mixed PET waste depolymerization using crude cutinase

Authors: Virender Kumar, Reinhard Wimmer, Cristiano Varrone *

…………………………………………………………………………………………………………………………...

The authors thank all the reviewers and the Academic Editor for diligently reviewing the manuscript and suggesting critical changes that have improved it. The changes made in the manuscript are highlighted in red color. A point-to-point response has been prepared to each concern/suggestion and is listed below:

Academic Editor

The paper is very interesting and deserves publication after revision according to the reviewers' comments.
I recommend specifying more details on the origin of PET powders, for example, industrial pellets, providing the commercial name, and expanding the section on the interaction of microplastic with enzymes citing the most relevant and recent literature of the last two years.

Response: The authors are thankful for the positive comments. The details on the origin of powders have been revised and provided in Table 1 (highlighted in red). The suggested changes have been made and new recent literature has been added.

Reviewer#1

Overall comments

This manuscript develops a bioprocess for the depolymerization of mixed PET waste using crude enzymes expressed extracellularly in E. coli. The enzymes FastPETase, LCC, and LCCICCG were selected to depolymerize amorphous PET powder and films of different sizes and crystallinity. After investigating the best conditions for depolymerizing crystalline PET and PET waste samples, it was observed that the yield of terephthalic acid in the hydrolytic process was maximum for LCCICCG, followed by LCC and FastPETase. Finally, extracellular E. coli cells producing LCCICCG were grown using a minimal medium supplemented with 0.1% ammonium chloride and 1% glycerol as nitrogen and carbon sources in a bioreactor with a final protein content and specific activity of 119±5 mg L-1 and 1232±18 U mg-1, respectively. Almost complete depolymerization of 13 g L-1 PET and 23.8 g L-1 post-consumer PET was achieved in 50 h using the crude LCCICCG supernatant, without enzymatic purification, at 62 ℃. A bioprocess was thus developed to depolymerize 100 g L-1 of mixed PET trays and bottle waste (MW1 and MW2), achieving 78% and 50% yield at 62 ℃ with a crude enzyme load of 2.32 mg g-1 PET in 60 h.

The work is indeed scientifically rigorous and of great importance in the field, especially with regard to obtaining more sustainable processes and the use of waste such as PET. However, some suggestions and corrections need to be made:

Response: The authors are thankful to the reviewer for his insightful and critical comments and highlighting the work's novelty. Please see below where we have addressed the concerns and suggestions.

  1. - Introduction section

The authors need to talk about chemical methods of PET hydrolysis and their disadvantages, as well as attempts to hydrolyze PET with other hydrolases, such as lipases, cutinases and esterases, listing their disadvantages compared to PETases;

Response: We welcome the suggestion of the reviewer. The relevant sentences have been added to the introduction section and highlighted in red. Please see Page 2 Line 46-52 and 67-74.

  1. - Methodology

Table 1: Can the authors provide any explanation for the choice of PET powder particle sizes applied in this work?

Response: Over the past years, the plastic recycling field especially enzymatic recycling has grown so much, and potentially good enzymes for PET degradation have been reported, and even applied in industrial processes. The purpose of this study was to compare the efficiency of the best enzymes (for PET depolymerization using PET substrates of different forms (powder and film of different sizes) and crystallinity. In this case, the idea was to use powder of different sizes to see its effect on the reaction using different enzymes. Smaller powder sizes increase surface area and reduce diffusion limitations, thereby improving enzymatic hydrolysis.

  1. Results and Discussion

Item 3.1 - line 278: the authors state that After expression and purification, a final yield of 26, 35, and 32 mg L-1 of culture was obtained. However, the authors do not express the specific activities of each enzyme. Provide.

Response: The specific activities are now mentioned Page 7 (lines 297-298 and 315-318) and provided in the supplementary Table S1.

  1. Item 3.2 - line 293: “Preliminary tests were done, using the purified enzyme (0.5 mg g-1PET), 13 g L-1 of PET film or powder, and 50 mM bicin NaOH buffer at the respective optimum temperature of the enzyme i.e. 50, 65, and 72℃ for FastPETase, LCC and LCCICCG  in 1.0 mL reactions.” The authors describe that they carried out initial PET degradation tests with these enzymes and that the optimum temperature for each was found. What was the main response variable in arriving at these temperatures?

Response: Thank you for asking this question. The main response variable in optimizing the reaction temperature was the depolymerization percentage (weight loss) and monomers (TPA, MHET) produced during the reaction. The results of the temperature optimization are provided in a supplementary file (Figure S3)

  1. line 313: The authors start a discussion about the influence of PET crystallinity and enzyme access on polymer degradation, as a way of explaining the differences found in Figure 2. Can't enzymatic activity also have an influence? It is important to note that the authors need to demonstrate the quantities of enzymes used in specific activity, since the mass is conserved in the case of denaturation, but the activity decreases.

Response: We agree with the reviewer’s observation that enzymatic activity can influence the overall depolymerization. We also agree that it is important to demonstrate the quantities of the enzyme as a specific activity.  It is to be noted that the specific activities of the enzymes were calculated using p-nitrophenyl butyrate assay and these units were maintained during the depolymerization assay. However, it is a standard practice to express the enzyme concentration as mg/g polymer to better evaluate the depolymerization potential of the enzymes with other studies. Therefore, the enzyme activities are only expressed in mg/g PET. But to address the reviewer’s concern, in section 3.2 the units are expressed as U/mg of protein used. It now says ‘The specific activities of the enzymes were calculated using pNPB assay and expressed as U/mg and an equal number of units were added to the reaction mixture in all depolymerization assays. For example, 968±15, 1430±27, and 2187±19 U mg-1 were used for FastPETase, LCC, and LCCICCG respectively’. The specific activities are now mentioned Page 7 (lines 297-298 and 315-318) and provided in the supplementary Table S1.

  1. Item 3.3 - Optimization of different parameters for enzymatic PET hydrolysis (It's listed as 2.3, but it should be 3.3, as well as the next item - check): the authors can't say that the system was optimized, because they didn't plan the experiments, but rather checked the variables influencing the system.

Response: Thank you for drawing our attention to this point. The list is now corrected. The section heading is now changed to ‘Different variables influencing enzymatic PET hydrolysis’. Page 9 Line 356

  1. - Figure 3 has very low resolution and it is difficult to see the error bars.

Response: The Figure 3 is now revised with high resolution.

  1. - Line 361: Also PET powder depolymerization was most efficient in 50 mM KH2PO4-NaOH buffer (pH 8.0 and 9.0) for FastPETase (22%), LCC (31%) and LCCICCG (58%). There was a rapid decline in depolymerization efficiency as the pH dropped below 7.0 due to the release of acidic hydrolysis products (TPA and MHET) which decreased the reaction mixture's pH. Did the authors verify the optimum pH for enzyme activity? This information is important for maintaining catalytic efficiency in the system. I reiterate that the authors need to express the enzyme quantity data in specific activity in the manuscript.

Response: Thank you for the query. The optimum buffer and respective pH of the reaction were verified and are presented in Fig. S2. It is to be noted that the specific activities of the enzymes were calculated using p-nitrophenyl butyrate assay and these units were maintained during the depolymerization assay. However, as mentioned above, it is a standard practice to express the enzyme concentration as mg/g polymer to better evaluate the depolymerization potential of the enzymes with other studies. Therefore, the enzyme activities are only expressed in mg/g PET. But to address the reviewer’s concern, in section 3.2 the units are expressed as U/mg of protein used. It now says ‘The specific activities of the enzymes were calculated using pNPB assay and expressed as U/mg and an equal number of units were added to the reaction mixture in all depolymerization assays. For example, 968±15, 1430±27, and 2187±19 U mg-1 were used for FastPETase, LCC, and LCCICCG respectively’. The specific activities are now mentioned at Page 7 (lines 297-298 and 315-318) and provided in the supplementary Table S1.

9.- Line 365: Interestingly, LCCICCG catalyzed PET hydrolysis produced more TPA than MHET compared to LCC and FastPETase in the buffer conditions tested. To what do the authors attribute this ability?

Response: Thank you for the question. LCCICCG seems to have better MHET hydrolysis activity, better substrate binding, and optimized enzyme thermal stability under the tested buffer conditions that contributed to the increased TPA yield compared to LCC and FastPETase. The initial rate of reaction and high thermal stability are crucial factors and are also evident in this case. Please see Page 8 Line 327-330

  1. Line 421: In the PET hydrolytic process, pH is a crucial factor with a huge impact on the overall depolymerization. Please provide further explanations.

Response: It has been provided and now reads as follows: ‘In the PET hydrolytic process, pH is a crucial factor with a huge impact on the overall depolymerization. The pH significantly affects enzyme stability, activity, substrate binding, and product solubility. The efficiency of PET hydrolases depends on the optimal pH conditions that support catalytic activity and structural integrity. Page 11 Line 453-455.

11.- Figure 7: were no replicates made? Provide erros bars

Response: The replicates were made of the analysis. The figure is now revised with error bars.

Reviewer #2

I have gone through the article and found it interesting for the scientific audience. The Authors have carried out an analysis of the literature, discussing ways to PET waste depolymerization using using FastPETase, LCC, and LCCICCG enzyme. Autor’s have shown comprehensive descriptions of the use of the enzymes for the different waste PET source used in a depolymerization process. Description of the research methodology is correct. The conclusions summarizing the conducted research are comprehensive. The work is interesting, easy to read, good and robust.

 Response: The authors thank the reviewer for his insightful and encouraging comments. As suggested, the additional details are provided in the manuscript and the response to the reviewer’s concerns can be found below.

Specific comments:

  1. There are some issues authors should consider: Novelty and strength of the work should be clearly presented in relation to previous published by authors and literature ones. Drawbacks and weakness must be also discussed. Strengthen the novelty of the work at the end of the introduction section. The novelty of a study should be clearly highlighted, to underscore its unique contributions to the field.

 Response: We thank the reviewer for his suggestion. The novelty and strength of the work have been highlighted. Drawbacks and weaknesses of the current process are also discussed along with future directions. The introduction section has been revised. Please see Page 2-3 Line 95-110 and Page 15 Line 562-571 and 583-588.

  1. Please strengthen the literature review and expand on how this work builds upon or differs from existing research, emphasizing the unique contribution. Compare effectiveness with other depolymerization technologies.

Response: The new literature is added. A comparative table is provided as Table S4 comparing different recent research studies on enzymatic depolymerization.

  1. Whole part at pages 14 and 15, lines 495-518 should be improved. Especially argumentations based on NMR results. Proton NMR of TPA: is there any peak at higher δ for TPA? Detail assignments should be presented. More detail about samples preparation for NMR “Samples were prepared by mixing 500 μL of supernatant with 25 μL (200 mM) of 3-(Trimethylsilyl) -2,2,3,3-tetradeuteropropionic acid sodium salt (TSP-D4) in D2O.”

Response: We thank the reviewer for the suggestion. The suggested part of the section is now improved especially the NMR results. Please see Page 14 Line 534-542.

Proton NMR of TPA: there are no other signals of TPA at higher chemical shifts. Carboxylic acid protons cannot be observed since the spectra were recorded at pH 7.85 where TPA is deprotonated. We have prepared a figure for the Supplementary Materials (Figure S7) showing the assignment of resonances in more detail. We do not quite understand which details are missing in the sample preparation section? Our description should be sufficient to allow the reproduction of our work, in that we exactly explained how the NMR samples were prepared.  We have changed the sentence slightly to : “Samples were prepared by mixing 500 μL of supernatant from the enzymatic reaction with 25 μL of a 200 mM solution of 3-(Trimethylsilyl) -2,2,3,3-tetradeuteropropionic acid sodium salt (TSP-D4) in D2O. The methyl signal of TSP served as an internal standard for the chemical shift (δ=0 ppm).” Please instruct, in case we misunderstood.

  1. Please explain “However, it can be negated using higher concentrations of NaOH (to form TPA salt) and ensuring that TPA salt concentrations do not go beyond the solubility limits.” in relation to obtained spectra and sample preparation method. Why authors did not present 13C NMR spectra?

Response: The effect of NaOH addition and formation of TPA has been explained. Please see Page 14 Line 549-557.

13C-NMR spectra were not recorded on the samples of this project. 13C-NMR is inherently insensitive and low concentrations would make them very time-consuming to run. We acknowledge that 13C-NMR can serve to corroborate the resonance assignments, however, we have in a previous work recorded 13C-NMR and 1H NMR (and 2D-NMR) of all compounds (Jamie-Azuara et al, 2023), so that we are confident about the assignments.

Reviewer 2 Report

Comments and Suggestions for Authors

Efficient bioprocess for mixed PET waste depolymerization using crude cutinase

Manuscript No.: polymers-3496310

I have gone through the article and found it interesting for the scientific audience. The Authors have carried out an analysis of the literature, discussing ways to PET waste depolymerization using using FastPETase, LCC, and LCCICCG enzyme. Autor’s have shown comprehensive descriptions of the use of the enzymes for the different waste PET source used in a depolymerization process. Description of the research methodology is correct. The conclusions summarizing the conducted research are comprehensive. The work is interesting, easy to read, good and robust.

There are some issues authors should consider:

  1. Novelty and strength of the work should be clearly presented in relation to previous published by authors and literature ones. Drawbacks and weakness must be also discussed. Strengthen the novelty of the work at the end of the introduction section. The novelty of a study should be clearly highlighted, to underscore its unique contributions to the field.

  1. Please strengthen the literature review and expand on how this work builds upon or differs from existing research, emphasizing the unique contribution. Compare effectiveness with other depolymerization technologies.
  2. Whole part at pages 14 and 15, lines 495-518 should be improved. Especially argumentations based on NMR results. Proton NMR of TPA: is there any peak at higher δ for TPA? Detail assignments should be presented. More detail about samples preparation for NMR “Samples were prepared by mixing 500 μL of supernatant with 25 μL (200 mM) of 3-245 (Trimethylsilyl) -2,2,3,3-tetradeuteropropionic acid sodium salt (TSP-D4) in D2O.”

Please explain “However, it can be negated using higher concentrations of NaOH (to form TPA salt) and ensuring that TPA salt concentrations do not go beyond the solubility limits.” in relation to obtained spectra and sample preparation method. Why authors did not present 13C NMR spectra?

Comments on the Quality of English Language

minor English imporvement

Author Response

(The authors gave the same response as above.)

Round 2

Reviewer 1 Report

Comments and Suggestions for Authors

Dear authors,
After reading the answers to the questions and carefully analyzing the manuscript, I could see that the questions have been fully taken into account, and that the manuscript has improved considerably in coherence, cohesion, becoming even more interesting and important for the field. In my opinion, the paper is ready to be accepted in this journal. 

Reviewer 2 Report

Comments and Suggestions for Authors

Authors responded well